# Reconstructing paleoclimate fields using online data assimilation with a linear inverse model

Walter A. Perkins[1] and Gregory J. Hakim[1]

[1]University of Washington, Seattle, WA, USA

*Correspondence to:* Walter Perkins (wperkins@uw.edu)

**Abstract.** We examine the skill of a new approach to climate field reconstructions (CFRs) using an online paleoclimate data assimilation (PDA) method. Several recent studies have foregone climate model forecasts during assimilation due to the computational expense of running coupled global climate models (CGCMs), and the relatively low skill of these forecasts on longer timescales. Here we greatly diminish the computational cost by employing an empirical forecast model (linear inverse model; LIM), which has been shown to have comparable skill to CGCMs for forecasting annual-to-decadal surface temperature anomalies. We reconstruct annual-average 2m air temperature over the instrumental period (1850 - 2000) using proxy records from the Pages 2k Consortium phase 1 database; proxy models for estimating proxy observations are calibrated on GISTEMP surface temperature analyses. We compare results for LIMs calibrated on observational (Berkeley Earth), reanalysis (20th Century Reanalysis), and CMIP5 climate model (CCSM4 and MPI) data relative to a control offline reconstruction method. Generally, we find that the usage of LIM forecasts for online PDA increases reconstruction agreement with the instrumental record for both spatial fields and global mean temperature (GMT). Specifically, the coefficient of efficiency (CE) skill metric for detrended GMT increases by an average of 57% over the offline benchmark. LIM experiments display a common pattern of skill improvement in the spatial fields over northern hemisphere land areas and in the high-latitude North Atlantic – Barents Sea corridor. Experiments for non-CGCM-calibrated LIMs reveal region-specific reductions in spatial skill compared to the offline control, likely due to aspects of the LIM calibration process. Overall, the CGCM-calibrated LIMs have the best performance when considering both spatial fields and GMT. A comparison with the persistence forecast experiment suggests that improvements are associated with the linear dynamical constraints of the forecast, and not simply persistence of temperature anomalies.

## 1 Introduction

Climate field reconstructions (CFRs) aim to provide essential information on climate variability beyond the instrumental record. These experiments take noisy and sparse proxies (e.g. tree rings, ice cores, isotope ratio measurements, etc.) and use them to infer a spatial estimate of relevant climate variables. A common approach to CFR uses a statistical regression model calibrated on the instrumental record to project as far into the past as data will allow (e.g. Mann et al., 1998, 2009; Smerdon et al., 2011b, see Smerdon et al., 2011a for a discussion and comparison of methods). These techniques provide a useful estimate of past spatial patterns (Wahl and Smerdon, 2012), but also have inherent limitations. For example, regression-based CFRs assume

climate state to be a function of the proxy data, which can lead to an underestimation of past climate anomaly amplitudes (Smerdon et al., 2011b; Wahl and Smerdon, 2012). Furthermore, because regression methods produce past spatial fields through combinations of primary variability modes (i.e. empirical orthogonal functions; EOFs), the resulting field is not guaranteed to be a physically consistent solution.

An alternate method of performing CFRs known as paleoclimate data assimilation (PDA) can circumvent some of the limitations inherent to regression-based methods. PDA broadly characterizes a set of techniques where observational information from proxy data is combined with dynamical information from climate models. Recently, the ensemble Kalman filter (EnKF) was adapted for use with time-averaged observations like those used in CFRs (Dirren and Hakim, 2005; Huntley and Hakim, 2010). Studies using the EnKF method and idealized psuedoproxy experiments have shown that it operates well under sparse
data availability (Bhend et al., 2012), and outperforms modern statistical CFR methods (Steiger et al., 2014). More recently, EnKF PDA was tested with real proxy data in the Last Millennium Reanalysis project (LMR; Hakim et al., 2016), and shows promising skill in reconstructing robust spatial fields in a computationally efficient manner. Due to the expense of performing coupled global climate model (CGCM) simulations and relatively low forecast skill, the initial EnKF adaptation for PDA does not use a forecast and instead reconstructs each time period independently using climatological data. This is known as an "of-
fline" approach. The EnKF method is traditionally accompanied by forward model forecasts to translate information between analysis time periods (e.g. reanalysis products of the instrumental era). Dynamical constraints from these forecasts can increase physical consistency and reconstruction skill given that the model has sufficient predictability on proxy timescales (e.g. Pendergrass et al., 2012). For CFR applications, predictability on seasonal and longer timescales is required. Ocean memory can be leveraged for inerannual (e.g. El Niño Southern Oscillation; ENSO) to potentially decadal predictability (Branstator
et al., 2012). However, at this timescale coupled climate models only seem to capture linearly predictable dynamics (Newman, 2013).

    Online assimilation has been attempted using other PDA techniques. Crespin et al. (2009) used forecasts from an earth system model of intermediate complexity (EMIC) in conjunction with the ensemble selection PDA method (see Goosse et al., 2006, 2010) to reconstruct surface temperatures, but did not investigate a comparison with an offline method. Annan and Har-
greaves (2012) performed a psuedoproxy experiment using a weighted ensemble selection method and a persistence forecast to reconstruct surface temperatures, but found no benefit compared to their offline experiments. Matsikaris et al. (2015) took a similar approach to Crespin et al. (2009), but used an ensemble of decadal forecasts from a coarse resolution CGCM instead of an EMIC. The authors found that the use of CGCM forecasts had skill, but it was not discernibly superior to the offline method. Possible reasons for the lack of improvement include low skill for regional decadal forecasts of temperature, and issues related
to ocean initialization for each decadal interval.

    These results suggest that neither the simple persistence forecast, nor a small ensemble of decadal CGCM forecasts add significant information to CFRs. In order to test the viability of a more traditional EnKF method we require the ability to perform annual forecasts for longer time spans (the past millennium) and in large ensembles (~100 members). These requirements rule out the use of a CGCM. Instead, we explore a simple, empirically-based forecast from a linear inverse model (LIM; Penland
and Sardeshmukh, 1995). A LIM encodes the linear dynamical properties of a system and produces forecasts that are subject

to the constraints of its derived linear modes. The forecast skill of LIMs is such that they are currently used for operational ENSO forecasts (Newman et al., 2009). Moreover, recent studies show LIM skill to be comparable to that of CGCMs when performing annual-to-decadal hindcast experiments over the instrumental era (Newman, 2013; Huddart et al., 2016).

Here, we propose a computationally efficient "online" data assimilation approach for use in paleoclimate field reconstruc-
5 tions. The primary goal is to investigate whether the addition of dynamical constraints with a forecast in the online case can increase reconstruction skill relative to the offline EnKF method, which has no forecasting. We perform a series of reconstruction experiments using annual forecasts from a cost-efficient LIM. Global average and spatial reconstructions from the online experiments are compared to results from both a persistence forecast method and the offline method of Hakim et al. (2016). In Section 2 we discuss the basics of the EnKF method and define the use of LIM forecasts in reconstruction experiments. Section
3 details the datasets used and the general experimental configuration. Section 4 discusses and compares results between the online and offline reconstructions, followed by conclusions in Section 5.

## 2  Online PDA

The Last Millennium Reanalysis (LMR) framework (Hakim et al., 2016) provides a setting to run many computationally efficient realizations of an offline climate reconstruction. Here we begin with it as the basis for our implementation and inves-
15 tigation of online PDA. Central to the LMR framework is the use of the ensemble Kalman filter (EnKF; Kalnay, 2003), which assumes gaussian distributed errors. The EnKF update equation (Eq. 1) describes the calculation of a posterior (analysis) state vector $\mathbf{x}_a$ through the optimal update to a prior (background) state vector $\mathbf{x}_b$ using proxy information,

$$\mathbf{x_a} = \mathbf{x_b} + \mathbf{K}[\mathbf{y} - \mathcal{H}(\mathbf{x_b})]. \tag{1}$$

The innovation, $[\mathbf{y} - \mathcal{H}(\mathbf{x}_b)]$, characterizes new information content as a difference between proxy observations in vector $\mathbf{y}$ and
20 observations estimated from the prior by $\mathcal{H}(\mathbf{x}_b)$ (hereafter denoted as $\mathbf{y}_e$). $\mathcal{H}()$ is a potentially non-linear operator that maps the prior state into observation space. The Kalman gain matrix $\mathbf{K}$, defined by Eq. (2), spreads information into the analysis weighted by prior covariance and the observational error covariance matrix $\mathbf{R}$

$$\mathbf{K} = cov(\mathbf{x}_b, \mathbf{y}_e)[cov(\mathbf{y}_e, \mathbf{y}_e) + \mathbf{R}]^{-1} \tag{2}$$

where $cov(a, b)$ represents a covariance expectation. The LMR framework uses a variant of the EnKF update, known as an
25 ensemble square root filter (EnSRF; Whitaker and Hamill (2002)). This process updates the ensemble mean and perturbations from the mean separately allowing for the serial assimilation of proxy data, and simplification of the update calculations.

Typical implementations of the EnKF method include a model forecast between analysis times. As stated earlier, the computational expense and low skill of CGCM forecasts prompted the use of the offline method where each year is reconstructed

independently without forecasting. Here, instead of using static prior ($\mathbf{x}_b$) at the beginning of each reconstruction year, the current year's posterior analysis is forecast forward by one year with a LIM defined by

$$\mathbf{x}_b^f = \mathbf{G}_1 \mathbf{x}_a. \tag{3}$$

The term $\mathbf{G}_1$ is a mapping term calculated from the calibration of a LIM that maps the current state to a forecasted state 1-year

later. Details of the EnKF reconstruction algorithm can be found in Appendix A. The formulation of the LIM used here is described in the following section.

## 2.1 Linear inverse model formulation

The linear inverse model (LIM; e.g. Penland and Sardeshmukh, 1995) used in this study closely follows the implementation described in Newman (2013). The basic equation describes a linearized dynamical system

$$\frac{d\mathbf{x}}{dt} = \mathbf{L}\mathbf{x} + \xi \tag{4}$$

as the tendency of an anomaly state vector $\mathbf{x}$, given by a dynamical operator $\mathbf{L}$, which is linearized about a mean state, plus random white noise, $\xi$. The dynamical operator $\mathbf{L}$ is assumed to be constant in time. After integrating (Eq. 4) in time, the solution is a mapping of $\mathbf{x}$ at time $t$ (in years) to a state at time $t + 1$

$$\mathbf{x}(t+1) = \mathbf{G}_1 \mathbf{x}(t) + \sigma(t) \tag{5}$$

where $\mathbf{G}_1$ is equivalent to $e^{\mathbf{L}}$. As in Newman (2013), we choose to empirically estimate $\mathbf{G}_1$ rather than $\mathbf{L}$ due to sampling deficiencies of a few highly damped eigenmodes of $\mathbf{L}$ on an annual timescale. Each LIM is constructed using an EOF basis retaining the leading 8 modes of variability. See Appendix B for a summary of the details associated with the $\mathbf{G}_1$ calculation.

While the simplicity of a LIM makes it well suited for the current application, it also has issues to be considered. First, LIM forecasts are performed using an EOF basis derived from the calibration data. The EOF basis is used to maximize the

variance captured by the fewest degrees of freedom. Although the dominant modes of variability can change in time during the reconstruction period, the space spanned by the variability cannot. This means that variability that falls outside the span of the EOFs will not be resolved. A second issue is that the LIMs tend to be damped (modes of L decay in time), which reduces the ensemble variance; we elaborate on this issue in the next section.

## 2.2 Ensemble calibration

In any ensemble forecast setting, a basic assumption is that the sample of ensemble members gives a good approximation to the statistics of the full system (Murphy, 1988). Sampling error often results in too-small variance, which can cause "filter divergence" where observational information is underweighted relative to the forecast prior and the ensemble variance collapses

toward zero. The online PDA technique presented here is especially vulnerable to filter divergence because all eigenmodes of $\mathbf{G}_1$ are damped (negative real eigenvalues). Moreover, the conversion of the analysis ($\mathbf{x_a}$) into EOF space at each timestep removes any spatial information that does not project upon the retained modes of a given LIM. Consequently, LIM forecasts lose ensemble variance in time.

5     There are a variety of well-tested methods available to address information loss in the forecast ensemble. Here we use an adaptation of the hybrid ensemble Kalman filter–3D variational scheme (Hamill and Snyder, 2000) to prevent filter divergence and to facilitate comparison with the offline PDA technique. This technique handles the loss of ensemble variance in the forecast ensemble ($\mathbf{x}_b^f$) by blending it with a static source ($\mathbf{x}_b^s$), which is the same climatological prior that is used independently for each year in the offline method. As a result, the update equations use a blended prior state $\hat{\mathbf{x}}_b^f$ (Eq. 6) and a blended Kalman 10   gain term $\hat{\mathbf{K}}$ (Eq. 7):

$$\hat{\mathbf{x}}_b^f = a\mathbf{x}_b^f + (1 - a)\mathbf{x}_b^s \tag{6}$$

$$\hat{\mathbf{K}} = \frac{(a)cov(\hat{\mathbf{x}}_b^f, \hat{\mathbf{y}}_e^f) + (1 - a)cov(\mathbf{x}_b^s, \mathbf{y}_e^s)}{(a)cov(\hat{\mathbf{y}}_e^f, \hat{\mathbf{y}}_e^f) + (1 - a)cov(\mathbf{y}_e^s, \mathbf{y}_e^s) + \mathbf{R}}. \tag{7}$$

Appendix A provides details on how this is incorporated into the reconstruction algorithm.

    In these hybrid DA equations, the parameter $a$ controls the relative weighting between static and forecast information 15   sources. When $a = 0.0$, reconstructions are identical to the offline case wherein the prior $\hat{\mathbf{x}}_b^f$ is reset to the static prior for every year with no blending. When $a = 1.0$, only forecast information is used with no contribution from static information.

## 3   Data and experimental configuration

The relative forecast skill of a LIM is dependent on the data used to empirically derive the mapping term $\mathbf{G}_1$. For this reason, we explore LIMs calibrated on four different data sets. CGCM calibration data are used from two last-millennium climate 20   simulations in the Coupled Model Intercomparison Project phase 5 (CMIP5; Taylor et al., 2012): the Community Climate System Model v4 (CCSM4; Landrum et al., 2013) and the Max Planck Insitute Earth System Model paleo-mode (MPI). These simulations cover a 1000 year pre-industrial (850–1850 C.E.) time period including volcanic forcing events (aerosols and greenhouse gases), solar variability, and human-related land cover changes. The 20th Century Reanalysis (20CR; Compo et al., 2011), a DA synthesis of observations and a weather forecast model, provides over 150 years of reanalysis data spanning the 25   instrumental record (1850–2012). Finally, we use the Berekely Earth surface temperature dataset (BE; Rohde et al., 2013) for observational calibration. BE provides a 65-year sample (1960–2014) with nearly complete global coverage. The different LIM calibration datasets used here span linear modes of predictability derived from model space to that of observations.

    The basic configuration we use for all experiments, including the offline control, involves a choice of data to sample as the static prior ensemble, an instrumental data source to calibrate proxy observation models, and a proxy record dataset. For 30   the static prior, we use annually-averaged 2m air temperature anomalies from the CCSM4 last-millennium simulation. The observation models for proxy data are calibrated against the NASA Goddard Institute for Space Studies surface temperature

analysis dataset (GISTEMP; Hansen et al., 2010) by linearly regressing the proxy timeseries against the nearest grid point in the calibration data (for details, see Hakim et al., 2016). All experiments use annually-resolved proxy records from the PAGES 2k Consortium (2013) database. These proxies have been ascertained to covary regionally with temperature and include: tree rings, ice cores, corals, sediment cores, and speleothems. Only proxies with a minimum of 10 years overlapping with the observation model calibration data, and a minimum calibration-fit correlation of 0.2 are used. It should be noted that the correlation threshold is not strictly necessary, but Hakim et al. (2016) found that this threshold did not quantitatively affect the reconstruction results. Here, the reduction in proxies to those with more information helps reduce computational costs, allowing a larger number of reconstruction experiments.

We reconstruct annual-mean 2m air temperature anomalies for the period of 1850–2000 C.E. as in Hakim et al. (2016). Using the four LIM calibrations, we search the parameter space of $0 \leq a \leq 1$ for a weighting between static and forecast information sources that improves the reconstruction compared to the offline method. We judge improvements by means of our chosen skill metrics (CE, correlation, and CRPS) for both GMT and spatial fields. Additionally, we perform a persistence experiment for comparison against LIM-based performance where the posterior for year $t$ is used as the prior for year $t+1$. The persistence forecast uses the same hybrid PDA blending scheme as the LIM forecast experiments to mitigate the effects of reductions in ensemble variance from the assimilation process. We account for the sensitivity to the proxy data used in a CFR through random resampling of available proxy data and the static prior ensemble. A single realization uses a random sample of 75% of the usable proxy records, and a 100 member sample of anomaly states from the prior. A total of 100 realizations are performed for each LIM calibration and blending coefficient. In order to make the realizations consistent between the experiments using different blending coefficients, we ensure the same sequences of random samples are taken by seeding the random number generator for each $a$-value. In total, this gives $10^4$ reconstructions of the climate state for each experiment. These reconstructions are then averaged to give the final analysis. See Section S1 in the supplementary information for a brief discussion of the computational costs associated with these experiments.

## 3.1 Skill metrics

The primary skill metrics used are correlation and the coefficient of efficiency (CE; Eq. 8,9; Nash and Sutcliffe, 1970). Correlation gives an overall sense of signal timing (phase), while CE is a stricter metric that is sensitive to signal timing, amplitude, and bias. Using these metrics, we compare the reconstructed ensemble-mean[1] values $x$ against GISTEMP validation values $v$. The value $\tau$ represents the number of validation times available (in this case representing the GISTEMP timespan of 1880 - 2000), an overbar (e.g. $\bar{v}$) denotes a temporal average, while $\sigma_x$ and $\sigma_v$ are the standard deviations of the respective time series. Skill scores are compared for the reconstructed global mean temperature (GMT) and spatial grid points.

---

[1]Ensemble mean represents the average taken over all ensemble members and all realizations.

$$corr = \frac{1}{\tau} \sum_{t=1}^{\tau} \frac{(x_t - \bar{x})(v_t - \bar{v})}{\sigma_x \sigma_v} \tag{8}$$

$$CE = 1 - \frac{\sum_{t=1}^{\tau} (v_t - x_t)^2}{\sum_{t=1}^{\tau} (v_t - \bar{v})^2} \tag{9}$$

We also use the continuous ranked probability score (CRPS; Gneiting and Raftery, 2007) as a comparison against CE skill metrics.

$$CRPS = \sum_{t=1}^{\tau} \left( \frac{1}{K} \sum_{i}^{K} \left| x_t^{(i)} - v_t \right| - \frac{1}{2K^2} \sum_{i}^{K} \sum_{j}^{K} \left| x_t^{(i)} - x_t^{(j)} \right| \right) \tag{10}$$

The CRPS is considered to be a 'proper' scoring technique which prevents manipulations of the data from overestimating the reconstruction skill; the measure reflects the mean absolute error and narrowness of the ensemble distribution. We use the CRPS as defined in Tipton et al. (2016) to calculate the CRPS of the reconstructed GMT for each realization (Eq. 10) where $x_t^{(i)}$ denotes a single member of a $K$-member ensemble. We take the score as the average CRPS over all realizations and use a Kolmogorov-Smirnov test on the resulting distribution to determine whether it is significantly different than the offline case with 95% confidence. As defined here, lower values of CRPS indicate better performance with a limiting case of CRPS = 0 being a perfect ensemble fit (no error and no ensemble spread).

## 4 Results and discussion

### 4.1 Validation of global mean temperature

Figure 1 displays global mean 2m air temperature (GMT) results validated against GISTEMP for all tested values of the blending parameter $a$. Every case except for the persistence forecast method yields CE values greater than the offline case. Correlations are higher than the offline benchmark for all experiments, including the persistence forecast. Best skill is achieved between the $a$-values of 0.7 to 0.9 with a steep drop in validation skill as $a$ approaches unity (a pure LIM forecast) due to filter divergence (the ensemble variance decreases with time, decreasing the weight on the proxies, so that the reconstructed states diverge from reality). The CCSM4 and 20CR LIM display the best overall CE performance with a 9% improvement over the offline method. These two experiments also display a 2% increase in correlation and have slightly smaller correlation than the persistence forecast experiment (Table 1). Figure 2 shows the reconstructed GMT from each experiment at the best blending coefficient (with respect to CE) compared to GISTEMP and the offline case. As evidenced by the high skill scores, the reconstructions capture the variability in the global temperature signal of GISTEMP quite well. Compared to the offline experiment, the forecasting experiments tend to decrease the amplitude of the interannual variability, which is at times largely overestimated by the offline experiment, and they tend to change the overall warming trend. There is no apparent systematic bias in the reconstructed GMTs, so skill changes for the different experiments are likely controlled by these two factors. Figure

3 displays the bootstrap estimate of 95% confidence bounds for reconstructed GMT CE, correlation, and trends at the same blending coefficients as chosen in Figure 2. For these blending choices, the CE scores for all LIM experiments show significant increases compared to the offline CE, while the persistence, 20CR, CCSM4, and MPI experiments show significant increases in correlation. Additionally, reconstructed GMT trends fall within the 95% confidence interval of the GISTEMP trend, except for the persistence case. The LIM experiments show trends much closer to the mean estimated GISTEMP trend than the offline case. CRPS values across blending coefficients (Fig. 4) show similar skill behavior as the CE metric (albeit mirrored in the vertical[2]). Specifically, GMT validation with CRPS shows that all LIM experiments outperform the offline method, with the CCSM4 and 20CR LIMs again having the best performace (18% better than the offline case). All LIM experiments' best CRPS scores are also significantly better than the offline case with 95% confidence. By and large, CRPS is giving nearly equivalent information as CE about the skill of the GMT reconstruction. However, as CRPS is a different metric, there are slight differences in results for the MPI and BE LIM cases when comparing CE and CRPS: the blending coefficient achieving the best score shifts to the next highest $a$-value in both cases, and the BE LIM outperforms the MPI LIM when considering CRPS (Table 1).

Both the CE and CRPS measures for different blending coefficients are affected by the degree of fit to the warming trend in the GISTEMP reference. The trends for all experiments are shown in Fig. 5. The trend of the offline case ($a = 0$) is 0.62 K/100 yrs, about 0.07 K/100 yrs above the GISTEMP trend and near the edge of the GISTEMP trend 95% confidence interval (Figure 3). However, for the MPI and CCSM4 LIM experiments, as well as the persistence forecast experiment, the reconstructed trend increases as the blending parameter $a$ increases. This increase in trend away from the GISTEMP trend for the MPI and CCSM4 experiments is reflected in the lowered CE (Fig. 1) and CRPS (Fig. 4) for the $a$-values from approximately 0.0 to 0.6. The reconstructed trend from the two CGCM-based LIM experiments begins to decrease around $a = 0.6$ where CE also shows a significant increase towards maximum values. The persistence forecast trend has the largest disagreement, increasing to approximately 0.72 K/100 yrs for $a = 0.9$, which results in the CE and CRPS never surpassing the offline benchmark in this case. The 20CR LIM only increases the reconstructed trend slightly over the GISTEMP trend for middle $a$-values. The BE experiment has a decreasing trend for increasing $a$, and drops to a very low trend of 0.38 K/100 yrs when $a = 0.9$. Interestingly, though the trends for the MPI, CCSM4, and 20CR experiments are below that of the GISTEMP trend for $a = 0.95$, their skill still outperforms the offline case in all three metrics. Despite the mismatch in the overall trend, these online forecasting methods still produce better matches of phase and amplitude of GMT variability for the reconstructed anomalies compared to the offline case.

The trend results also illustrate the relative amount of proxy data utilization between these different experiments. Given that every experiment uses the same prescribed list of seeds to generate proxy record samples, the differences in reconstructed trend can only arise from differences in weighting of the proxies or the LIM forecasts. Since LIMs are calibrated on detrended data and their forecast modes are damped, the forecast contribution to a long-term global mean trend is likely small; the trend is instead governed by utilization of proxy information. For the EnKF PDA method, the weighting of information is controlled by the prior ensemble variance and proxy error variance. The proxy error variance is fixed for all experiments we perform,

---

[2]Best results for CRPS occur at minimum values instead of the maximum.

so the changes in the reconstructed trend are a result of how the LIM forecasts affect the ensemble variance. In all forecast experiments, skill and the reconstructed trends drop off severely as $a$ approaches 1.0. When using only forecast information ($a = 1.0$), the ensemble variance collapses due to the damped properties of the LIMs, which results in filter divergence. The BE LIM case reaches its maximum CRPS and CE values at smaller $a$ and also has the lowest reconstructed trends of the LIM experiments. This suggests the BE forecast produces less ensemble variance than the other LIMs, possibly due to forecast mode damping or poor projection of the posterior analysis into the LIM EOF space. The eigenvalues of the BE LIM's leading two forecast modes have e-folding times of 5.4 and 1.5 years, respectively. This is in the same range of the leading forecast modes of the CGCM-calibrated LIMs (e.g. 3.7 and 1.2 year e-folding times for the MPI LIM). Consequently, a poor projection of the analysis ensemble onto the forecast modes of the BE LIM is likely the cause of the reduced ensemble variance. The persistence forecast displays an interesting disparity between the skill metrics; overall, it performs the best in correlation, but the worst in CE and CRPS. Having the largest reconstructed trend suggests that the persistence case has the highest weighting of proxy data. With a persistence forecast there is no damping of reconstructed spatial anomalies or truncation of the ensemble variance from projection into EOF space. The resulting higher proxy weighting may explain why the persistence case correlation is better than the other forecasting methods. The linear observation models in each case are based on a calibration against GISTEMP, so proxies with a better calibration fit (higher correlation) with GISTEMP have less error variance, and therefore, have more influence on the posterior analysis. The persistence case allows more information from the influential (well-correlated) proxies into the analysis because the prior variance is larger. However, from the CE and CRPS values, which are sensitive to more than just signal phase matching, it is clear the general trend mismatch degrades the quality of the persistence reconstruction compared to the offline benchmark.

Removing the linear trend from each case allows for an examination of how well the reconstructions capture variability not associated with the warming trend (i.e. interannual and decadal variability; evident in Fig. 6). Generally, the performance increases for the detrended data over the offline case are much larger than for the full time series. Compared to validation with the full time series, the correlation of the detrended offline case drops from 0.9 to 0.67, and the CE drops from 0.77 to 0.29 (Table 2). With respect to CE, all experiments (including the persistence method) improve upon the offline benchmark. The 20CR LIM achieves the best improvement over the offline case (72% increase), while the persistence case shows the least improvement (35% increase). Except in the BE LIM experiment, detrended correlation metrics again increase slightly over the offline case. The BE LIM hovers around the benchmark correlation for $0.0 < a < 0.7$ and then drops below it. A CE improvement with no change in correlation implies that the BE LIM improves the detrended anomaly amplitudes and bias, but does not improve the signal timing compared to the offline case.

The CE and correlation skill metrics of the offline experiment both decrease when calculated on the detrended GMT. In contrast, the CRPS improves by 7% (minimizing from 12.5 to 11.6). CRPS rewards reduction in mean absolute error, and 'narrowness' of the forecast ensemble, whereas CE and correlation depend more on variance properties of the reconstructed time series. In removing the linear warming trend, we remove a large degree of the time series' variance and subsequently lose the associated skill in CE and correlation. In the case of CRPS, the linear trend is only a source of mean error when it does not closely match the reference trend. When we remove the trend, the mean errors decrease (the ensemble spread is unaffected)

and the CRPS metric improves. Figure 7 shows CRPS for all blending coefficients with detrended data. The behavior is quite similar to the full GMT CRPS, and as the detrended CE reflects, even the persistence forecast shows improvement over the offline method. An aspect that stands out with detrended CRPS is that the persistence forecast achieves the best value when $a = 1.0$. A cursory examination of the detrended GMT timeseries of the persistence case compared to the detrended GISTEMP GMT timeseries (see supplement; Figure S1) reveals that it captures some decadal variability over the instrumental record, but none of the interannual variability; i.e., the $a = 1.0$ persistence reconstruction gives a smoothed representation of the GMT. This again highlights a difference between the two metrics of CE and CRPS. The CRPS metric, which generalizes to the mean absolute error of the ensemble summed over time, does not penalize the smoothed GMT signal approximately bisecting the interannual signal for $a = 1.0$. The CE metric, which sums the squared errors of the ensemble mean and then normalizes by the climatological variance, does penalize this behavior.

## 4.2 Validation of spatial fields

Here we compare the skill of the spatial fields with the offline case using correlation and CE; CRPS is omitted because the full spatial field ensembles are too large to store. For ease of visualization, we provide CE difference maps to highlight changes relative to the offline case, but full spatial skill maps can be found in the supplementary material (Figures S2 and S3). Over the globe, skill is mostly positive, but there are a few regions with highly negative skill departures such as the Southern Hemisphere oceans and the high-latitude North Atlantic to Barents Sea corridor (Fig. 8). In the high-latitude Atlantic region, the proximity to the sea ice edge seems to negatively affect the ability to constrain the temperature field when using only proxies. All LIM-forecasting cases show improvements to CE in the same North Atlantic to Barents Sea area, and across northern Europe into Asia; there are also smaller skill increases across western North America. Studies of LIM predictability have shown the North Atlantic/Barents sea region can have forecast skill on annual and longer timescales (e.g. Hawkins and Sutton, 2009; Newman, 2013). This may be one reason why skill increases are common in this area for all LIM experiments. An inspection of grid point temperature values northeast of Iceland (see supplement; Figure S5) shows that the reconstructed temperature variance is largely overestimated in the offline case, and that there is a slight trend in the reconstruction that is not present in GISTEMP data. The CE increase for the CCSM4 LIM experiment (increasing from -7.2 to -1.7) relates to a temperature variance reduction by approximately 70% compared to the offline case. The CCSM4, MPI, and BE LIMs generally show large CE increases in the high-latitude southern ocean. In contrast, the reconstruction with the 20CR LIM does not improve the southern ocean at all and has large deficiencies over many ocean areas. The persistence case generally shows decreases in CE across large areas of the globe. Of the global mean CE for each grid, the CCSM4 LIM gives the best performance, increasing the global mean CE by 0.09, followed by the MPI LIM with an improvement of 0.06. The 20CR and persistence cases show decreases in average spatial skill across the grid, with the 20CR being worst with a global mean CE change of $-0.18$. The BE LIM, while showing improvements over North Hemisphere land areas, has compensating decreases in skill over the ocean that make the global mean CE nearly equivalent to the offline case. All global mean CE values, except in the BE LIM case, are significantly different (at 95% confidence) from the offline case when comparing grid point skill distributions using a Student's t-test. Changes in spatial correlation (see supplement; Figure S4) are generally small in regions where CE increases, which suggests improvements are

not related to signal phasing. However, some of the large decreases in CE for the 20CR, BE, and persistence experiments do coincide with areas of correlation decreases.

In the spatial results, there is a clear distinction between LIMs calibrated on data from the shorter instrumental era (20CR and BE), and the millennium-scale climate simulation data (CCSM4 and MPI). Compared to the offline spatial skill, large areas of CE skill degradation are apparent for both the 20CR and BE LIM reconstructions. The 20CR LIM CE skill degredations are large in amplitude ($\Delta$CE$< -1$) and mostly over ocean regions. It is surprising that the 20CR LIM has the worst spatial skill given that it has the best GMT timeseries CE skill. However, previous CFR studies show similar results (Annan and Hargreaves, 2012; Wang et al., 2014) where the spatial averaging over a poorly reconstructed field can boost the signal-to-noise ratio for large-scale indices enough to result in positive index skill. The situation is somewhat different in this study. In the offline case, spatial skill is reasonably positive, but when adding 20CR LIM forecasts, the spatial CE skill decreases and the GMT CE skill increases. A possible interpretation for this behavior is that the degradation in spatial field fidelity is compensating for aspects of the GMT that are overestimated in other experiments. For example, the offline reconstructed GMT overestimates interannual variability during certain time periods when compared to GISTEMP, but the usage of LIM forecasts mitigates this effect. Compared to the CCSM4 LIM experiment, the reconstructed GMT for the 20CR LIM experiment reduces the sum of the squared error (numerator term in CE) by approximately 5%. The bias accounts for only about 1% of the total sum squared error term, which leaves the trend and anomaly amplitude agreement as primary candidates for the error reduction. As shown earlier, the 20CR GMT trends tend to track lower than the two CGCM LIM experiments and closer to the GISTEMP trend (Figure 5). The trend reduction in the 20CR experiment appears to be caused by large areas of negative trends in the Southern Hemisphere (see supplement; Figure S6).

The reason behind the poor spatial performance of the 20CR LIM appears to be linked to its EOF basis (Figure 9). The leading EOF of the 20CR experiment lacks the ENSO/PDO-like pattern of the other LIMs and instead focuses on variability structures in the southern ocean. This is a region of high variability due to its proximity to the storm track, but there are also fewer pressure observations available for assimilation by the 20CR (see Figure 3 in Woodruff et al., 2011), especially during the early portion of the record. Many of the large decreases in CE are located in these same regions, which leads us to speculate that the features of the 20CR LIM may be influenced by artifacts in the 20CR dataset. Another consideration for the lower performance of both instrumental era LIMs is that they are based on shorter records that coincide with variability related to anthropogenic forcing. The BE LIM displays a primary EOF more similar to the CGCM-based LIMs, but still has skill problems over large ocean areas. Separating the global warming trend from the LIM by means of linear detrending is bound to leave residual signals that affect LIM forecast modes. A LIM based on a shorter record may not have enough of a sample to properly characterize representative modes of variability over a longer time span. While the BE LIM is based on 65 years of data, it produces much less spatial skill degradation than the 150 years of 20CR data. This suggests there may be confounding factors influencing the skill in the 20CR experiment between the length of record, linear detrending, and the assimilation method used to create the 20CR data.

## 5 Conclusions

We have outlined and tested a new method for performing online paleoclimate data assimilation (PDA) for climate field recon­structions (CFRs) using linear inverse models (LIMs). We tested four different LIMs empirically derived from surface temper­ature data from the following data sets: Berkeley Earth (BE), the 20th Century Reanalysis (20CR), and two last-millennium
climate simulations (CCSM4 and MPI) from the Coupled Model Intercomparison Project phase 5 (CMIP5). We also performed a persistence forecast experiment for comparison. In general, we find that LIM-enabled online assimilation improves upon the offline results for both the global average and spatial field of 2m air temperature.

    Broadly speaking, the LIM experiments show good ability to reconstruct many aspects of the GISTEMP GMT data including interannual and low freqency variability. The largest skill improvements occur for skill metrics calculated on the detrended
GMT, which suggests the LIM forecasts are adding useful information at interannual timescales. The coefficient of efficiency (CE) for detrended metrics show an average increase around 57%, while correlations increase around 4%. The continuous ranked probability score (CRPS) metrics increase by an average of 15% across all LIM experiments. Skill metrics tend to maximize for blending coefficients with a higher weighting on LIM forecast information ($0.7 < a < 0.95$). Spatial skill reveals that the addition of LIM forecasting provides spatial information in regions where the offline method performs poorly—
including North Hemisphere land areas, and the North Atlantic to Barents Sea corridor. Large skill improvements seen in the North Atlantic through Barents Sea region are primarily a result of better constraining the temperature variance at these locations. The two LIMs calibrated on instrumental era data (20CR and BE) display large regions over the ocean where the skill degrades compared to the offline case. Even with the large areas of improvement, the 20CR LIM decreases the area-weighted average CE ($-0.18$), and the BE LIM area-weighted average breaks even. In contrast, the two CGCM-based LIM experiments
show area-weighted average CE increases of 0.09 (CCSM4) and 0.06 (MPI), respectively. When considering both GMT and spatial skill results, the CGCM-based LIMs have the best overall performance with the CCSM4 LIM slightly outperforming the MPI LIM. The persistence forecast fails to improve the more stringent GMT skill metrics (CE and CRPS) as well as general spatial skill, but does well in GMT timeseries correlation. Subsequently, this suggests that the improvements of online reconstructions when using a LIM are due to forecast information and not simply the addition of temporal persistence.

Though we are reconstructing instrumental era surface temperatures, it is an interesting result that CGCM-calibrated LIMs based on last millennium (850-1850) simulations have the best overall performance. This could mean that having long-running samples of variability that do not contend with influences from anthropogenic forcing are beneficial for reconstruction purposes. The LIMs used in these experiments were all calibrated on data with the least-squares linear fit trend removed. In order for LIMs based on observational data sources to achieve similar results, it may be necessary to employ a more sophisticated method
of filtering out the global warming signal. However, one benefit of using CGCM-based LIMs is that it enables forecasts for a much larger set of climate-related quantities than are available from observations alone.

    In this work, we have shown that we can improve both GMT and spatial field skill over the offline EnKF PDA method through the inclusion of a simple forecast model. A previous comparison of offline and online PDA using a CGCM as a forecast model found no discernible difference in reconstruction skill (Matsikaris et al., 2015), and earlier studies of the EnKF

PDA method forewent the usage of forward models citing insufficient model skill to justify the expense (Bhend et al., 2012; Steiger et al., 2014). Our results show that an online method can increase reconstruction fidelity, and more importantly that it can be done using an empirical forecasting method that is nearly as computationally efficient as the offline approach. As such, this method provides a useful foundation for further investigation of incorporating dynamical constraints of a forecast model

into climate field reconstructions.

## Appendix A: Online assimilation algorithm

This section details the data assimilation equations used to perform paleoclimate field reconstructions, and the algorithm steps for a single realization of an online climate reconstruction.

### A1 Ensemble square root filter (EnSRF)

The EnSRF approach (Whitaker and Hamill, 2002) uses an ensemble sampling approach to solve the Kalman filter equations by separating the ensemble mean ($\bar{\mathbf{z}}_b$) and ensemble perturbations about that mean ($\mathbf{z}_b' = \mathbf{z}_b - \bar{\mathbf{z}}_b$). Note that $\mathbf{z}_b$ represents an augmented state vector, $\mathbf{z}_b = \begin{bmatrix} \mathbf{x}_b \\ \mathbf{y}_e \end{bmatrix}$, combining the prior state ($\mathbf{x}_b$) and the estimated observations ($\mathbf{y}_e$). The EnSRF method allows for the serial assimilation of proxy observations using the equations

$$\bar{\mathbf{z}}_a = \bar{\mathbf{z}}_b + \mathbf{K}[y_i - \bar{\mathbf{y}}_{ei}] \tag{A1}$$

$$\mathbf{z}_a' = \mathbf{z}_b' + \tilde{\mathbf{K}}\mathbf{y}_{ei}' \tag{A2}$$

for each proxy $i = 1, ..., p$. The mean state $\bar{\mathbf{z}}_b$ is an $m \times 1$ column vector, the ensemble perturbations $\mathbf{z}_b'$ is an $m \times n$ matrix, the mean estimated observation for proxy $i$, $\bar{\mathbf{y}}_{ei}$, is a scalar value, and perturbations about the mean estimated observation $\mathbf{y}_{ei}'$ is a $1 \times n$ row vector. Note that when observations are serially assimilated, the Kalman gain (Eq. 2) simplifies to

$$\mathbf{K} = \frac{cov(\mathbf{z}_b', \mathbf{y}_{ei}')}{var(\mathbf{y}_{ei}') + \sigma_i^2} \tag{A3}$$

where $\sigma_i^2$ is the observational error for proxy $y_i$ and the denominator is now a scalar value . The perturbation update equation $\tilde{\mathbf{K}}$ is given by

$$\tilde{\mathbf{K}} = \left[ 1 + \sqrt{\frac{\sigma_i^2}{var(\mathbf{y}_{ei}') + \sigma_i^2}} \right]^{-1} \mathbf{K} \tag{A4}$$

Finally, to adapt the hybrid assimilation scheme into the EnSRF method, we incorporate the data source blending as shown in Eq. (7). At this point, the blended state ($\hat{\mathbf{z}}_b^f$) contains both flow dependent and static information. After incorporation, the Kalman gain (Eq. A3) becomes

$$\hat{\mathbf{K}} = \frac{(a)cov(\hat{\mathbf{z}}_b^{f\prime}, \hat{\mathbf{y}}_{ei}^{f\prime}) + (1-a)cov(\mathbf{z}_b^{s\prime}, \mathbf{y}_{ei}^{s\prime})}{(a)var(\hat{\mathbf{y}}_{ei}^{f\prime}) + (1-a)var(\mathbf{y}_{ei}^{s\prime}) + \sigma_i^2},$$

(A5)

the perturbation Kalman gain (Eq. A4) becomes

$$\tilde{\mathbf{K}} = \left[1 + \sqrt{\frac{\sigma_i^2}{(a)var(\hat{\mathbf{y}}_{ei}^{f\prime}) + (1-a)\mathbf{y}_{ei}^{s\prime} + \sigma_i^2}}\right]^{-1} \mathbf{K},$$

(A6)

and the mean and perturbation updates from Eq. (A1) and Eq. (A2) become

$$\bar{\mathbf{z}}_a = \bar{\hat{\mathbf{z}}}_b^f + \hat{\mathbf{K}}[y_i - \bar{\hat{\mathbf{y}}}_{ei}^f]$$

(A7)

$$\mathbf{z}_a' = \hat{\mathbf{z}}_b^{f\prime} + \tilde{\mathbf{K}}\hat{\mathbf{y}}_{ei}^{f\prime}.$$

(A8)

## A2   Assimilation algorithm

1. Choose a static ensemble prior ($\mathbf{x}_b^s$) of $n$ members, and group of $p$ proxies ($\mathbf{y}$) to assimilate.

2. Calibrate observation models for each proxy record by applying a univariate linear fit against co-located instrumental data.

3. Create an estimated observation ensemble ($\mathbf{y}_e^s$) for each proxy record using their corresponding observation model and augment the prior ensemble to form the static state ensemble, $\mathbf{z}_b^s = \begin{bmatrix} \mathbf{x}_b^s \\ \mathbf{y}_e^s \end{bmatrix}$. This an $(m+p) \times n$ state vector that will be updated during assimilation.

4. For each reconstruction year:

   (a) If not the first reconstruction year, then reset $\mathbf{z}_b^s$ back to the original static prior and form a blended prior ($\hat{\mathbf{z}}_b^f$) using Eq. (6).

   (b) For each proxy $y_i$ from $\mathbf{y} = [y_1, y_2, ..., y_p]$:

      i. If the current proxy has no observations for the current year, then skip to the next proxy

      ii. Else, select the matching estimated observations ($\hat{\mathbf{y}}_{ei}^f$, $\mathbf{y}_{ei}^s$) corresponding to the current proxy $y_i$ from the augmented states ($\hat{\mathbf{z}}_b^f$, $\mathbf{z}_b^s$).

      iii. Calculate the mean and perturbation of the estimated observations and augmented state vectors for use in the serial ensemble square root filter method.

    iv. Use the perturbations of $\hat{\mathbf{z}}_b^f$, $\mathbf{z}_b^s$, $\hat{\mathbf{y}}_{ei}^f$, $\mathbf{y}_{ei}^s$ to form a blended Kalman gain as shown in Eq. (A5) and Eq. (A6).

    v. Update the mean and perturbations of $\hat{\mathbf{z}}_b^f$ and $\mathbf{z}_b^s$ by using the blended gain matrices from step iv in Eq. (A7) and Eq. (A8).

    vi. Reassemble $\mathbf{z}_a$ and $\mathbf{z}_a^s$ by adding the ensemble mean back into the perturbations. These will be used for the next proxy assimilated as $\hat{\mathbf{z}}_b^f$ and $\mathbf{z}_b^s$.

(c) After all proxies have been assimilated, extract the climate field $\mathbf{x}_a$ from the augmented analysis state $\mathbf{z}_a$.

(d) Perform a LIM forecast on $\mathbf{x}_a$ using Eq. (3) resulting in $\mathbf{x}_b^f$.

(e) Recalculate the estimated observations $\mathbf{y}_e^f$ from $\mathbf{x}_b^f$ and augment the state to form $\mathbf{z}_b^f$.

(f) Return to step 4.(a).

## Appendix B: LIM calibration

The following steps are performed to empirically derive a LIM from a given data source. The steps detail how we find the mapping term $\mathbf{G}_1$ shown in Eq. (5).

1. If the calibration data contains seasonal signals (e.g., monthly data), then they are removed by smoothing data with a 1-year running mean.

2. The data is converted into anomaly format by removing the climatolical mean for each individual month.

3. The resulting anomaly is detrended. This removes a large degree of the skill found by Newman (2013) when using instrumental data, but we are focused on forecasting modes of interannual variability, not the secular warming trend.

4. The detrended anomaly data is projected into EOF space where the leading 8 modes of variability are retained. The number of modes retained encompasses those with $e$-folding times (decorrelation time scales) near 1-year or greater based on analysis of $\mathbf{G}_1$ (as calulated in the following step) in these experiments. The near-1-year threshold was chosen because forecast modes with $e$-folding times much less than 1-year have a small impact on annual forecasts due to the relatively quick signal decay.

5. Finally, we determine $\mathbf{G}_1$ based on the lag-covariance statistics of the calibration data. Specifically, we solve $\mathbf{C}(1) = \mathbf{G}_1\mathbf{C}(0)$ is solved for $\mathbf{G}_1$ $\mathbf{C}(1) = \left\langle \mathbf{x}(t+1)\mathbf{x}(t)^T \right\rangle$.

*Acknowledgements.* This paper represents a portion of the first-author's Master's Thesis at the University of Washington. This research has been supported by awards from the National Science Foundation (awards AGS-1304263 and AGS-1602223), and the National Oceanic and Atmospheric Administration (NA14OAR4310176). The first author has also been supported by the National Science Foundation Graduate Research Fellowship under Grant No. DGE-1256082. The authors would like to thank Matthew Newman for his input on constructing

linear inverse models, and Robert Tardif for his large contributions and many thoughtful discussions related to software development for this project.

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

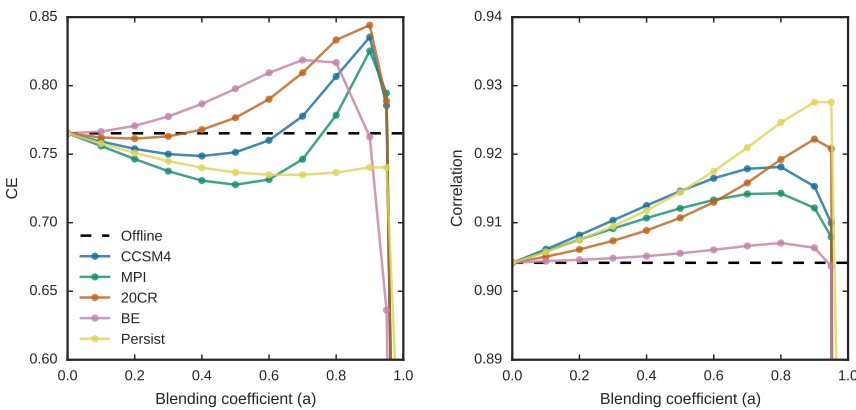

**Figure 1.** Comparison of global mean 2m air temperature coefficient of efficiency (CE; left) and correlation (right) metrics for different blending coefficients. The colored lines represent the different LIM calibration experiments using data from the Community Climate System Model v4 (CCSM4), NOAA 20th Century Reanalysis v2 (20CR), Max Plank Institute Earth System Model (MPI), Berkeley Earth Surface Temperatures (BE); or the persistence forecast case (Persist). The offline benchmark is depicted as the horizontal dashed black line.

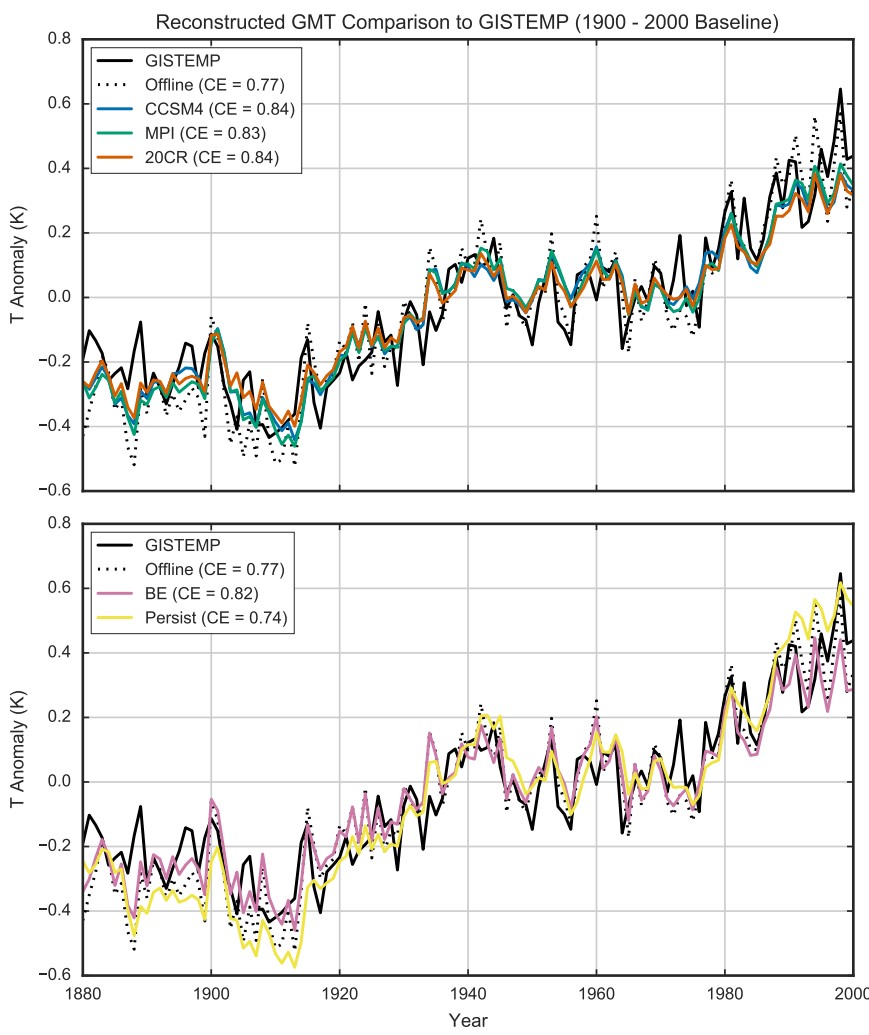

**Figure 2.** Reconstructed global mean 2m air temperature compared to GISTEMP (solid black) and the offline experiment (dotted black) for each online experiment. The GMT plotted for each forecasting experiment is from the blending coefficient that achieves the highest GMT CE skill (CCSM4: $a = 0.9$, MPI: $a = 0.9$, 20CR: $a = 0.9$, BE: $a = 0.7$) except for the persistence case where $a = 0.9$ was used. The top row shows the three forecasting experiments with the highest GMT CE score, while the bottom row shows the other two experiments.

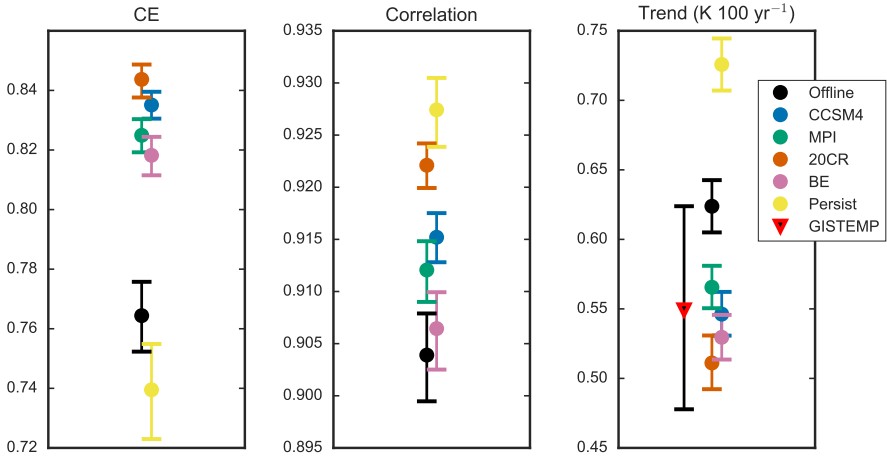

**Figure 3.** Bootstrap uncertainty estimates (95% confidence interval) for CE scores (left), correlation (middle), and GMT trends (right). The blending coefficient that achieves the highest GMT CE skill is shown for each experiment (CCSM4: $a = 0.9$, MPI: $a = 0.9$, 20CR: $a = 0.9$, BE: $a = 0.7$) except for the persistence case where $a = 0.9$ was used.

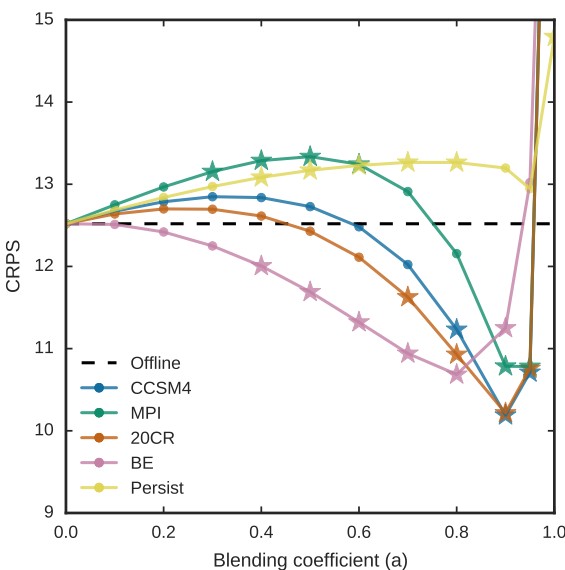

**Figure 4.** Comparison of global mean 2m air temperature continuous ranked probability score (CRPS) for different blending coefficients. The colored lines represent the different forecasting experiments, while the offline benchmark is depicted as the horizontal dashed black line. Starred points indicate a statistically significant (95% confidence) difference between the offline benchmark and online experiment.

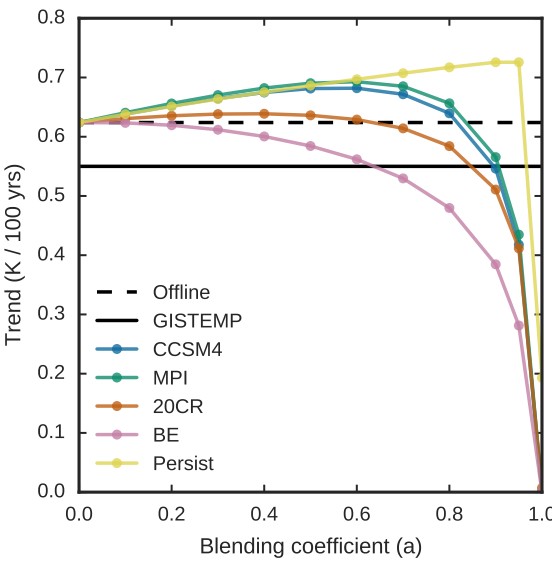

**Figure 5.** Calculated trends from a least squares fit against the reconstructed global mean 2m air temperature (1880 - 2000). Colored lines depict the calculated trends for each LIM experiment across a range of blending coefficients, while the black lines represent the benchmark trends calculated from the offline reconstruction (dashed) and GISTEMP data (solid).

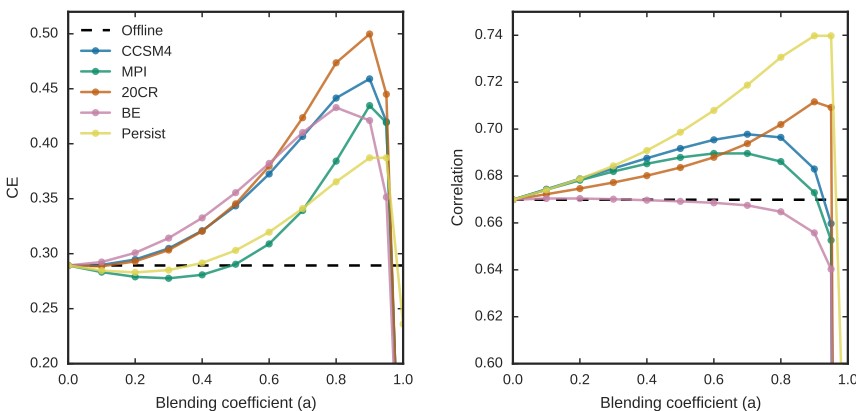

**Figure 6.** Comparison of detrended global mean 2m air temperature CE (left) and correlation (right) metrics across different blending coefficients for all experiments. The colored lines represent the different forecasting experiments, while the offline benchmark is depicted as the horizontal dashed black line.

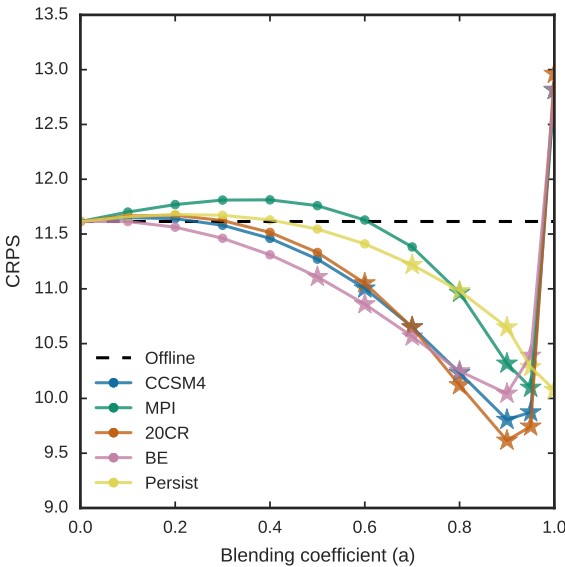

**Figure 7.** Comparison of detrended global mean 2m air temperature continuous ranked probability score (CRPS) for different blending coefficients. The colored lines represent the different forecasting experiments, while the offline benchmark is depicted as the horizontal dashed black line. Starred points indicate statistical significance (95% confidence) between the offline benchmark and online experiment.

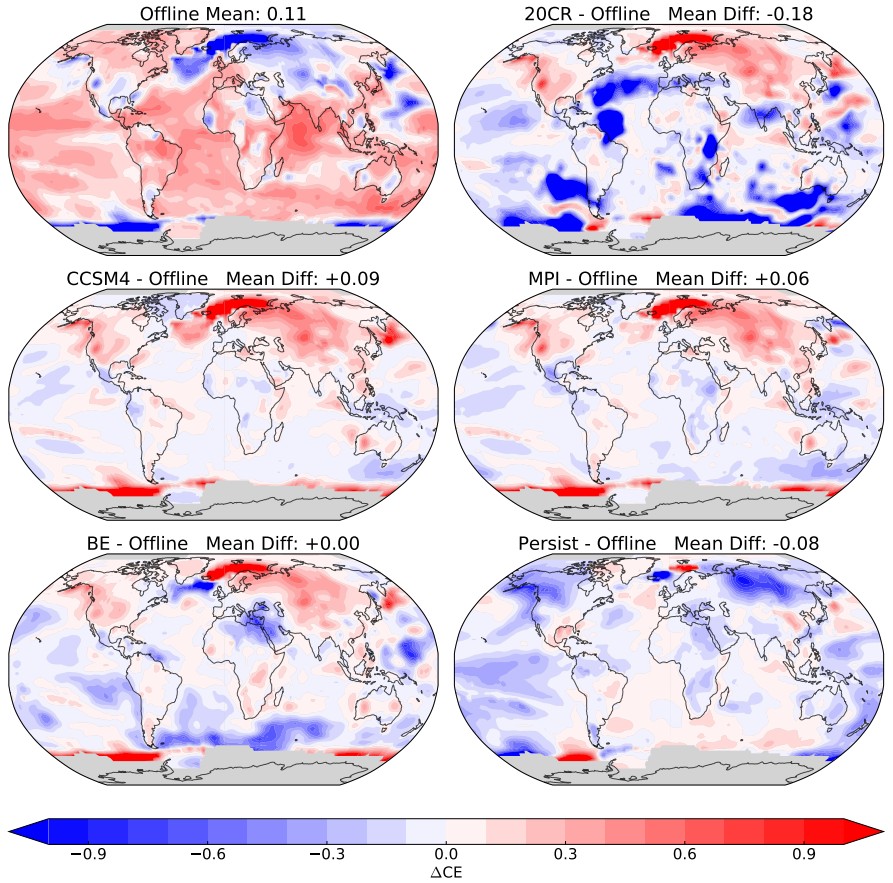

**Figure 8.** Spatial maps displaying the difference in coefficient of efficiency (CE) from the offline case. Difference maps are displayed for each forecasting experiment using the blending coefficient that achieves the highest full GMT CE skill (CCSM4: $a = 0.9$, MPI: $a = 0.9$, 20CR: $a = 0.9$, BE: $a = 0.7$) except for the persistence case where $a = 0.9$ was used. The reference CE of the offline case is shown in the upper left, and uses the same color scale as the difference maps. Area-weighted global average differences are given in the title of each panel. All global mean differences except in the BE LIM case are significantly different than the offline benchmark with 95% confidence.

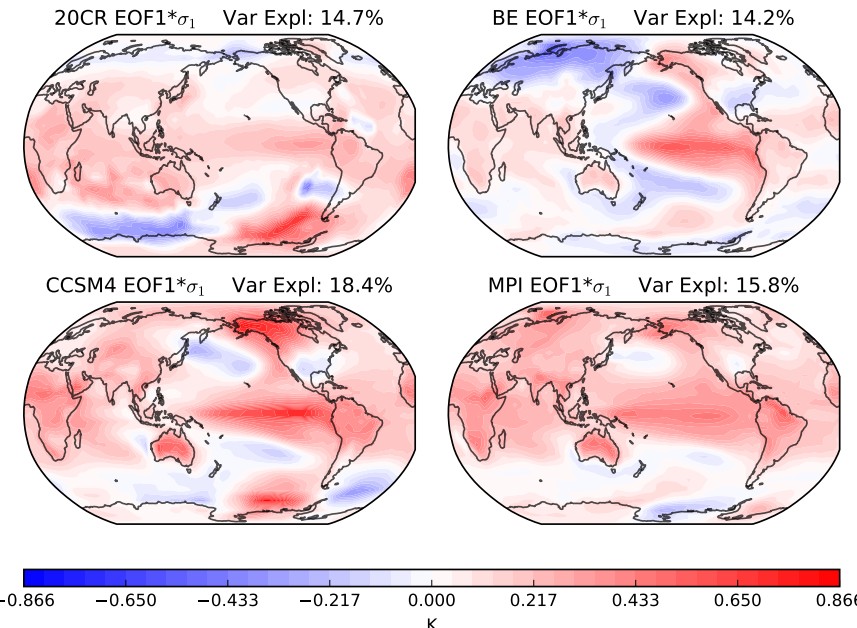

**Figure 9.** Leading empirical orthogonal function (EOF) from the basis for each LIM calibration. The total fraction of the variance explained is given in the title of each panel. All EOFs have been multiplied by their corresponding singular value.

**Table 1.** Best value of the coefficient of efficiency (CE), correlation (r), and continuous ranked probability score (CRPS) validation metrics for global mean 2m air temperature. For each experiment, best values are given with the corresponding blending coefficients (*a*) that achieved it and the percentage change compared to the offline case. A (*) indicates which experiment achieved the best performance in a given metric. Offline validation metrics are given for reference.

| Full GMT | Max CE | %$\triangle$CE | CE a-value | Max r | %$\triangle$r | r a-value | Min CRPS | %$\triangle$CRPS | CRPS a-value |
|---|---|---|---|---|---|---|---|---|---|
| Offline | 0.77 | | | 0.90 | | | 12.5 | | |
| Persist | 0.77 | 0 | 0.0 | *0.93 | 3 | 0.9 | 12.5 | 0 | 0.0 |
| BE | 0.82 | 7 | 0.7 | 0.91 | 1 | 0.8 | 10.7 | -14 | 0.8 |
| CCSM4 | *0.84 | 9 | 0.9 | 0.92 | 2 | 0.8 | *10.2 | -18 | 0.9 |
| 20CR | *0.84 | 9 | 0.9 | 0.92 | 2 | 0.9 | *10.2 | -18 | 0.9 |
| MPI | 0.83 | 8 | 0.9 | 0.91 | 1 | 0.8 | 10.8 | -14 | 0.95 |

**Table 2.** Best value of the CE, correlation, and CRPS validation metrics for detrended global mean 2m air temperature. For each experiment, best values are given with the corresponding blending coefficients ($a$) that achieved it and the percentage change compared to the offline case. A (*) indicates which experiment achieved the best performance in a given metric. Offline validation metrics are given for reference.

| Detrended GMT | Max CE | %△CE | CE a-value | Max r | %△r | r a-value | Min CRPS | %△CRPS | CRPS a-value |
|---|---|---|---|---|---|---|---|---|---|
| Offline | 0.29 | | | 0.67 | | | 11.6 | | |
| Persist | 0.39 | 35 | 0.9 | *0.74 | 11 | 0.9 | 10.1 | -13 | 1.0 |
| BE | 0.43 | 48 | 0.8 | 0.67 | 0 | 0.1 | 10.0 | -14 | 0.9 |
| CCSM4 | 0.46 | 59 | 0.9 | 0.70 | 5 | 0.7 | 9.8 | -16 | 0.9 |
| 20CR | *0.50 | 72 | 0.9 | 0.71 | 6 | 0.9 | *9.6 | -17 | 0.9 |
| MPI | 0.43 | 48 | 0.9 | 0.69 | 3 | 0.7 | 10.1 | -13 | 0.95 |