# Peer review of "Reconstructing paleoclimate fields using online data assimilation with a linear inverse model"

_Climate of the Past, 2016_

## Referee Comment (RC1) · Anonymous Referee #1 · 12 Jan 2017

Review on Reconstructing past climate by using proxy data and a linear climate model Walter A. Perkins and Gregory J. Hakim

I find that this article does not belong to the journal of Climate of the Past. Thus I reject the publication. However, I find the idea novel and interesting and I would suggest to submit a revised version to a more theoretical journal.

Here is why it doesn't belong to Climate of the Past: The idea of using LIM as a substitute for otherwise expensive online data assimilation sounds wonderful, and would prove to be it if one didn't need to introduce a parameter $a$. But as authors showed with $a$ being equal to 1, the results of linear online DA are worse than of offline DA. Then the question I pose to authors is how to choose an optimal $a$? What would be the criteria for optimal $a$? Looking at the tables 1 and 2 it seems only 20CR has highest CE, $r$, and

lowest CRPS at the same $a$, mainly 0.9. For other models highest CE is reached with one value of $a$, while highest $r$ with another value of $a$. Therefore an investigation of how to choose an optimal $a$ is lacking in this paper, and without an optimal $a$ I don't see how to extend this work to practical applications of the past climate state reconstruction. There is yet another manifestation of more theoretical work to be done, mainly in Sec. 4.2, where authors find inconsistency between the best model 20CR in terms of a scalar skill (CE, $r$, and CRPS) but worse in terms of spatial reconstruction. They propose that it could be due to a short time scale but this could be checked. And again, does this mean that $\alpha = 0.9$ is not optimal for 20CR? Therefore what I suggest is to study the methodology in a theoretical framework by revising the article and submitting it to a theoretical journal.

Other major comments: LIM is calibrated on model 1 (CCSM4) without any data assimilation over a period 850-1850, on model 2 (MPI) without any data assimilation over a period 850-1850, on model 3 with data assimilation (20CR) over a different period 1850-2012, and on a data set over yet a different time period (1950-2010 I would assume, though it is not mentioned in the paper). Thus the models are completely different in terms of the time period, use or not of the data, and only being the data. This makes it hard to compare and draw conclusions. Instead LIM should be calibrated on a model without DA, on the same model with DA, and on observations used in that DA.

As the prior authors used results of the CCSM4 model, the same model they used for LIM calibration. It appears that linear CCSM4 DA provides good results in terms of both scalar skills and spatial reconstruction. Is it because there is less inconsistency? How would it change if the prior was from another model?

In order to provide a fair comparison authors need to include "expensive" online DA (using a nonlinear model instead of LIM).

Minor comments: Page 7, Line 18: Why is there a shift in blending coefficient? This is

again related to my comment on how to choose an optimal $a$.

Page 7, Line 31: Why is there improvement compared to offline DA even though the trend is largely underestimated for $\alpha = 0.95$?

It would be interesting to introduce another metric – bias – in order to check whether the model either underestimates or overestimates the observed values.

I suggest plotting time series of averaged temperature of different models against observations for best $a$ for CE, for best $a$ for $r$, and for best $a$ for CRPS.
* * *

---

## Referee Comment (RC2) · Anonymous Referee #2 · 30 Jan 2017

Summary: Perkins and Hakim demonstrate climate field reconstruction using an "on-line" data assimilation technique. In their methodology, time slices of the reconstructed climate field are tied together with a Linear Inverse Model (LIM) approximation to a coupled climate model (which are themselves far too computationally expensive to assimilate proxy data in a climate reconstruction context). Perkins and Hakim implement the approach for estimating surface temperature, using LIMs calibrated on several different coupled climate models, and compare results to an offline approach. For certain choices of a blending parameter, the authors are able to achieve improvements in global mean temperature with all of the LIMs compared to the offline case. Changes in field reconstruction skill compared to the offline case are non-uniform across space, but tend to improve on average for some of the LIMs.

General Comments:

This paper is a very nice contribution to the field of paleoclimate reconstruction. Data assimilation approaches have been of interest to the paleoclimate community for a long time, and an "online" version represents a real advance- in my opinion this is much more satisfying than the offline reconstruction approaches. The authors have implemented a non-trivial experimental design, and it's a cherry on top that the new approach even yields improvements in skil over past methods.

The presentation of the work in the paper feels incomplete in several ways. The figures show comparisons of metrics of skill, and comparisons of estimated climate fields to a benchmark estimate, but no visualization of the reconstructions themselves, or comparisons to the actual target (the GISTEMP field and/or GMT time series). Though the interest of the authors is primarily on the increases in skill compared to previous reconstruction approaches, the paleoclimate community will want to see that estimates from this approach are reasonable compared to the actual target field. I appreciate that it's not reasonable to provide such figures for each of the many reconstruction experiments, but perhaps show some visualization of the GMT time series and error of reconstructed field relative to GISTEMP for one LIM experiment at the optimal value of the blending coefficient "a", and put the others in a supplementary document. (If the total number of figures is a concern, the authors might consider putting figures 1-3 into supplementary documentation and keeping just figures 4 and five in the main text, as all 5 are a bit redundant).

The authors have also neglected to describe of tabulate the computational expense associated with their reconstruction exercises. In addition, I wonder if they plan to make code for carrying out any of the reconstructions publicly available. Sharing code is perhaps the very best way to get other researchers to use and build on (and cite!) the advances in your work. One such place the authors might consider archiving their code would be the NCDC NOAA Paleoclimate Software Library: https://www.ncdc.noaa.gov/paleo/softlib/

Finally, more commentary on interpretation of the model and results are needed in the

discussion. In general I am a fan of combined results & discussion sections. In the present case though, the authors can provide much more "discussion" along with the delivery of the "results." Additional interpretation, and speculation as to reasons for observed results, will make the paper much richer, more interesting, and scientifically valuable.

Specific Comments:

Title: Suggest changing the title to assert what is novel about this paper: paleoclimate fields are reconstructed using an *online* data assimilation scheme. How about something along the lines of "Reconstructing paleoclimate fields with an online data assimilation methodology"? (Note also that "paleoclimate reconstruction" automatically implies proxy data are used.)

Abstract: pp 1, Line 5: LIMs have been shown to have comparable skill to CGCMs in what sense? This statement currently seems to vague. pp 1, Line 15-17: The last sentence may need to be revised or made more specific, to address the meaning of the "dynamical evolution" to which the authors attribute improvements in skill. When I think of "dynamics," I think of the description of the underlying physical mechanisms driving changes in time, where the term is used in contrast to a "statistical" description. The LIM is purely statistical though, so I think the authors mean the term in the sense of using the model forecast as a prior for each subsequent timestep.

Introduction: pp 1, Line 23: The approach to CFRs described constitutes a whole class of techniques, so change "This technique provides. . .. But also has inherent limitations" to "these techniques provide. . .. But also have. . ." pp 2, lines 1-3: It seems the physical consistency issue due to use of EOFs is also a limitation of the method presented in this paper though, right? Seems a bit disingenuous to list this here as if it's a limitation the present approach will address. pp 3, line 2: The authors might expand upon what they mean by "dynamical" at the first use of the word here, to make the precise nature of their contribution more immediately accessible to a wider audience. After reading the

paper thoroughly, I see they mean simply the use of a forecast from the LIM from the previously assimilated state as the prior for the next state. However on my first read, I thought they were claiming the use of physics-based information in the forecasts. line 5: Even a very quick (1-2 sentence) overview of the gist of the "offline method of Hakim et al. (2016)" would be useful here for the reader unfamiliar with this previous paper. Briefly summarize the difference between the online and offline approaches to be compared. pp 4, lines 16-19: Doesn't the implementation of the LIM in an EOF basis make this methodology subject to the same limitations as regression-based CFRs as described on pp 2, lines 1-4? Line 26: How many modes are retained in this study? (This detail is sufficiently important to be moved from the appendix to the main paper). What's the justification for the choice based on e-folding times of a year or greater? Are results sensitive to number of retained modes?

Data and experimental configuration: pp 5, line 24: "For the prior, we used . . .. the CCSM4 last-millennium simulation": Do the authors mean this is the model used for the climatological prior used for the blending used to prevent the collapse of the ensemble as described at the end of the previous section? If so: I would expect the EOFs of the prior and the CCSM4-based LIM to be the same, but different for the other CGCM-based LIMs, thereby perhaps giving the CCSM4-based LIM an advantage, or at least somehow controlling the divergence of that ensemble differently than for the three other CGCM-based LIMs? Line 25-26: The linear observation models for proxy data" should be described in enough detail to enable reproducibility. Are the proxy data simply linear in temperature of the gridcell containing each proxy location? Or a collection of gridcells representing the regional signal of each record referred to in the next sentence? Also, is there any particular justification for the choice of the GISTEMP product for calibrating the proxy models? Finally, it's interesting the authors use several proxy types with known differences in their spectral signatures. Is there any difference in the construction of linear observation models for the lower versus higher frequency proxies? Even if not in this work, the authors might acknowledge this issue and note it as an area to expand on in future work. pp 6, line 4: Optimal in what sense? What

is the criterion used to determine the optimum? Line 18 and forward: Consider using the term "validation" rather than "verification", here and throughout the paper (and in future climate reconstruction papers!) "Verification" is rooted in the latin word for truth. Of course, there is no ground truth to compare against in paleoclimate reconstruction, and estimates can only be shown to be valid in light of the known data and uncertainties, rather than true! Line 26, eqn 10: Provide a sentence or two of insight into how to interpret the CRPS statistic for readers unfamiliar with it. For example you can describe here where you introduce it that lower CRPS is better, or describe limiting cases of its value.

Results and Discussion

Pp 7, line 6: Verification → validation Line 10: provide interpretation of the steep drop in skill as the parameter a goes to one. Line 15: be more specific than writing the CRPS and CE results are "generally consistent." Do you mean the rank of models is the same as measured by both statistics? Line 18-20: Similar to preceding comment: it's imprecise to say there are "slight differences in results... when comparing CE and CRPS." These are two different metrics that measure different things in the first place. I wonder again if the authors mean to make a statement comparing the rank of models as measured by the two different metrics? In all figures, the authors show central estimates across ensembles, but no measures of uncertainty. Once it has been established in the results that the skill varies with the blending coefficient, it might be interesting to show some analysis of estimates and uncertainty across ensemble members for fixed "a" (probably at the value that optimizes one metric or another). For example, I'm curious about the spread in trend values across reconstruction ensemble members in figure 3 for fixed values of "a". I'm also curious how these compare to the uncertainty in estimates of the trend as measured by GISTEMP. Pp 7., line 31- pp. 8, line 3: I would speculate that this underestimation of trend in combination with skillful match of phase and amplitude of GMT variability might be interpreted in terms of the paleoclimate proxies as high-frequency bandpasses of the climate signal, that do not

tend to preserve the low-frequency signal. This is a well-known feature of many dendrochronologies, for example, although there do exist "standardization" methodologies to prepare tree ring time series to preserve the low-frequency signal. It would be interesting to know whether the proxy time series used in this study have been prepared using methods aiming to preserve low frequency climate variability.. Subsection 4.1 is missing figures and reporting of the estimated GMT time series compared to the target GMT time series. Pp. 9, line 21– seems odd not to show some spatial measures of skill against the target, rather than just against the offline case. Line 23-25: The authors should change "All LIM-forecasting cases show improvements to CE, most notably in the same North Atlantic to Barents Sea area" to "All LIM-forecasting cases show improvements to CE in the same North Atlantic to Barents Sea area." As written currently, this seems to falsely state that CE improves everywhere compared to the offline case for all LIM-forecasting reconstructions, rather than just in the region noted. Is there climatic significance to this North Atlantic/Barents Sea area that might explain why the LIM forecast- based reconstructions seem to improve skill there compared to the offline case? Or can the authors speculate as to why this region has low skill in the offline case to begin with to explain the near uniform improvements there under forecasting? Pp 10, line 3-4: "There is a clear distinction between LIMs calibrated on data from the shorter instrumental era, and the millennium-scale climate simulation data"– This is an interesting point. Remind readers explicitly at this point which are which, so that readers can easily reference what you're talking about in the figures. Also, can you describe the distinction you mean clearly and precisely? Looks to me like the millennial-scale ones have fewer regions of large-amplitude degradation in CE relative to the offline case.

Conclusions: Before stating conclusions in terms of improvements relative to the offline case, authors should state conclusions about the basic ability of the methodology to estimate the target. (Note this will require adding another set of analysis comparing reconstructions to GISTEMP target to the results.)

---

## Author Comment (AC1) · 1 Mar 2017

Walter A. Perkins and Gregory J. Hakim

wperkins@uw.edu

**1   Major Comments**

I find that this article does not belong to the journal of Climate of the Past. Thus, I reject the publication. However, I find the idea novel and interesting and I would suggest to submit a revised version to a more theoretical journal. Here is why it doesn't belong to Climate of the Past: The idea of using LIM as a substitute for otherwise expensive online data assimilation sounds wonderful, and would prove to be it if one didn't need to introduce a parameter $a$.  But as authors showed with a being equal to 1, the results of linear online DA are worse than of offline DA. Then the question I pose to authors is how to choose an optimal $a$?

**What would be the criteria for optimal $a$?**

We thank the referee for commenting on this paper. On this first point, we disagree with the assessment that this paper does not belong in the journal of Climate of the Past (CP), and our reply to that can be found below. On the need for a tuning parameter (such as $a$) in the usage of ensemble data assimilation methods, it is common due to the affect such parameters have on ensemble variance. As in Hamill and Snyder (2001), we present results for a range of the blending coefficient ($a$), but show that there is a general range of blending that gives improved validation metrics compared to the offline case. We do not give an absolute determination of an optimal ($a$) because, as we show in the paper, the choice of metric for determining the best value for ($a$) is subjective. The use of correlation, the coefficient of efficiency, and the continuous ranked probability score (CRPS) target different aspects of comparison between the reconstructed and reference data (though CE and CRPS are very similar). The details of the differences in different skill metrics are described in Section 3.1. Altogether, the determination of which metric is best depends on what the end user prioritizes.

**There is yet another manifestation of more theoretical work to be done, mainly in Sec. 4.2, where authors find inconsistency between the best model 20CR in terms of a scalar skill (CE, r, and CRPS) but worse in terms of spatial reconstruction. They propose that it could be due to a short time scale but this could be checked. And again, does this mean that $a = 0.9$ is not optimal for 20CR?**

We agree that this result can be more thoroughly discussed. The 20CR dataset is a re-analysis performed using surface pressure observations from 1850 to present (Compo et al. 2011). The observational coverage of the southern oceans, especially during the early portion of the record is very low (see Fig. 3 in Woodruff et al. 2010). This is a large region of high variability due to the southern hemisphere storm track. As Fig. 7 in our paper shows, the primary mode of spatial variability in the annually averaged 20CR dataset is quite different in character from the other 3 datasets used. The pattern of variability is focused on regions in the southern oceans, which is where we see
a lot of the CE skill degradation. This coupled with the knowledge that it is a region of high variability with few observations leads us to speculate that the features of the 20CR LIM may be influenced by artifacts of the 20CR dataset. Another instrumental reanalysis product spanning 1900 – 2012 (ERA-20C; Poli et al. 2016) has a similar first EOF and forecast mode to the 20CR. The discrepancy between the GMT and spatial performance for the 20CR experiment was surprising, but similar results were found in idealized pseudo-proxy experiments (Annan Hargreaves 2012, Wang et al., 2014). These studies show that spatial averaging can boost the signal-to-noise ratio for large-scale indices resulting in higher index skill despite poor spatial reconstruction performance. Here we have a situation that is slightly different. The spatial results for the offline case are decent, but the degradation of spatial skill by the 20CR LIM reconstruction enhances the skill in the GMT signal. We interpret this as the spatial degradation having a moderating effect on an aspect of the global signal that is overestimated in the offline case such as the interannual variance or the warming trend. We can provide a breakdown of this idea with plots of calculated spatial trends and interannual GMT variance over time. Again, $a = 0.9$ is the optimal blending when considering CE skill of GMT. Changing the blending parameter will not change the behavior of the forecasts, only how much information is used from the forecast. If a user was inclined to use the 20CR LIM for a reconstruction, they could optimize between spatial skill and GMT skill, but the use of 20CR LIM forecast information will only reduce spatial skill. From what we show in the paper, there are better options than the 20CR LIM to use for spatial reconstruction performance.

**Therefore, what I suggest is to study the methodology in a theoretical framework by revising the article and submitting it to a theoretical journal.**

Regarding the larger question of relevance to this journal, we believe that there exists well-established precedent for studies like the current one. The field of paleoclimate data assimilation (PDA) has had a number of recent theoretical advances, many of which were published in Climate of the Past (CP). For example, Crespin et al. (2009)

extend the ensemble selection DA method to perform online forecasts with an intermediate complexity climate model. However, they make no comparison with its offline predecessor. Bhend et al. (2012) use an idealized pseudo-proxy experiment to test an offline ensemble square root filter approach. In this work, they use a parameter known as covariance localization to prevent the collapse of ensemble variance, and they also discuss the extension to a coupled model forecast online method as the next step. Annan and Hargreaves (2012) perform an idealized pseudo-proxy experiment with an ensemble selection method to test different geographic densities of proxy observations. They also try a persistence forecast method as an online case, but find no benefits to the reconstruction from the persistence forecast in all cases. Matsikaris et al. (2015) perform a direct comparison between online and offline methods using the ensemble selection method with a 10-member ensemble forecast from a coupled GCM. They also find no discernible benefits to using the online method compared to the offline experiment. They follow up with a study (Matsikaris et al., 2016) further investigating why the online forecast method does not show improvements to their reconstruction targets. Thus it is clear that there are numerous papers in CP similar to ours; this is why we submitted the paper to CP. We have presented the only study to our knowledge that shows a viable online PDA method that can be used for long timescales with large ensembles while also documenting improvements over the offline alternative. Additionally, we document the performance of the method using real proxy data in real reconstructions. We plan to provide code and sample data along with the revised manuscript. The connection to previous CP literature and our novel results make this study both timely and relevant for this journal.

**LIM is calibrated on model 1 (CCSM4) without any data assimilation over a period 850-1850, on model 2 (MPI) without any data assimilation over a period 850-1850, on model 3 with data assimilation (20CR) over a different period 1850-2012, and on a data set over yet a different time period (1950-2010 I would assume, though it is not mentioned in the paper). Thus, the models are completely different in terms of the time period, use or not of the data, and only being the data. This**

**makes it hard to compare and draw conclusions. Instead LIM should be calibrated on a model without DA, on the same model with DA, and on observations used in that DA.**

That would be a nice framework for a future study, but it is clearly well beyond the scope of this paper. In any case, our method has sampled widely different sources for variability on which to calibrate the LIMs. During the instrumental period the separation between forced and natural variability is unclear. However, the last millennium simulations are a good substitute for a measure of long-term natural variability under weaker forcing regimes. As far as the use or not of observations in the calibration data, we will note that for climate models there are no common systems that provide data assimilation for simulations like this. That means a comparison like the one suggested would have to be done using systems oriented for reanalysis, which are very different from climate models. Our results show that reanalysis products may have inherent properties making them less useful for our application. Additionally, the observations used for assimilation during the instrumental period are not necessarily something that we can use as a LIM forecast model in reconstruction. For example, the 20CR assimilates pressure observations. Other reanalysis products use a suite of different measurements including satellite radiances and upper air measurements. These are not easily useable gridded products and are outside the scope of variables we would use in paleoclimate applications. As we said, these are topics that can be explored in future research. We will change the text to explicitly define that the BE dataset covers the years 1960-2014.

**As the prior authors used results of the CCSM4 model, the same model they used for LIM calibration. It appears that linear CCSM4 DA provides good results in terms of both scalar skills and spatial reconstruction. Is it because there is less inconsistency? How would it change if the prior was from another model?**

We did check results using the MPI last millennium simulation as our prior and the NOAA merged land-ocean surface temperature analysis (MLOST) as the calibration
data for the proxy observation models. The CCSM4 LIM still outperformed the MPI LIM by a similar margin in the spatial results (the CCSM4 spatial average CE was +.02 above the MPI case). The GMT skill results show the two models again at a virtual tie. This suggests that the CCSM4 LIM provides forecasts that generate results more consistent with the GISTEMP reference we use for validation.

**In order to provide a fair comparison authors need to include "expensive" online DA (using a nonlinear model instead of LIM).**

As stated in our introduction, the current computational costs of performing coupled model forecasts in large ensembles over long time periods make this suggestion impractical. Moreover, other studies have explored this possibility (with smaller ensembles forecasting on decadal timescales) and shown no benefit over offline DA. This aspect and the knowledge that a LIM can be comparable in forecast skill to the coupled models (Newman 2013) is why we chose to try a LIM for online paleoclimate data assimilation. We believe this is a fair baseline comparison showing the potential of an online method compared to an offline method. Despite the simplicity of this approach, we show improvements in reconstruction skill over the offline method, which is the first to our knowledge for any online technique. "Expensive" online PDA options likely won't be feasible with ensembles of the size we use here anytime soon.

**2   Minor Comments**

**Page 7, Line 18: Why is there a shift in blending coefficient? This is again related to my comment on how to choose an optimal a.**

There is a shift in blending coefficient between CE and CRPS because they are different skill metrics. CE is calculated based on mean squared error properties, while CRPS is calculated on accumulated mean absolute error. Though they are both sensitive to bias, phase, trend, and amplitude differences, there are no guarantees that they

will give the same results. This is especially apparent for detrended GMT skill of the persistence case when $a = 1.0$.

**Page 7, Line 31: Why is there improvement compared to offline DA even though the trend is largely underestimated for $\alpha$ = 0.95?**

There is improvement because the trend is not the only consideration of the CRPS/CE skill metrics. Other aspects (phase and amplitude) can still be improving while the trend difference is not large enough to decrease the skill measure.

**It would be interesting to introduce another metric – bias – in order to check whether the model either underestimates or overestimates the observed values.**

The CE metric can be separated to show skill related specifically to bias. We will consider expanding the GMT and spatial breakdown of CE skill between bias and other aspects of the reconstruction.

**I suggest plotting time series of averaged temperature of different models against observations for best a for CE, for best a for r, and for best a for CRPS.**

We agree that we should include a figure of the actual reconstructed GMT for selected reconstructions. Thank you for this suggestion.

**3   References**

Annan, J. D.,  Hargreaves, J. C. (2012). Identification of climatic state with limited proxy data. Climate of the Past, 8(4), 1141–1151. http://doi.org/10.5194/cp-8-1141-2012

Bhend, J., Franke, J., Folini, D., Wild, M.,  Brönnimann, S. (2012). An ensemble-based approach to climate reconstructions.  Climate of the Past, 8(3), 963–976. http://doi.org/10.5194/cp-8-963-2012

Compo, G. P., Whitaker, J. S., Sardeshmukh, P. D., Matsui, N., Allan, R. J., Yin, X., . . . Worley, S. J. (2011). The Twentieth Century Reanalysis Project. Quarterly Journal of the Royal Meteorological Society, 137(654), 1–28. http://doi.org/10.1002/qj.776

Crespin, E., Goosse, H., Fichefet, T., Mann, M. E. (2009). The 15th century Arctic warming in coupled model simulations with data assimilation. Climate of the Past, 5(3), 389–401. http://doi.org/10.5194/cp-5-389-2009

Hamill, T. M., Snyder, C. (2000). A Hybrid Ensemble Kalman Filter–3D Variational Analysis Scheme. Monthly Weather Review, 128(8), 2905–2919. http://doi.org/10.1175/1520-0493(2000)128<2905:AHEKFV>2.0.CO;2

Matsikaris, A., Widmann, M., Jungclaus, J. (2015). On-line and off-line data assimilation in palaeoclimatology: a case study. Climate of the Past, 11(1), 81–93. http://doi.org/10.5194/cp-11-81-2015

Matsikaris, A., Widmann, M., Jungclaus, J. (2016). Influence of proxy data uncertainty on data assimilation for the past climate. Climate of the Past, 12(7), 1555–1563. http://doi.org/10.5194/cp-12-1555-2016

Newman, M. (2013). An Empirical Benchmark for Decadal Forecasts of Global Surface Temperature Anomalies. Journal of Climate, 26(14), 5260–5269. http://doi.org/10.1175/JCLI-D-12-00590.1

Poli, P., Hersbach, H., Dee, D. P., Berrisford, P., Simmons, A. J., Vitart, F., . . . Fisher, M. (2016). ERA-20C: An atmospheric reanalysis of the twentieth century. Journal of Climate, 29(11), 4083–4097. http://doi.org/10.1175/JCLI-D-15-0556.1

Wang, J., Emile-Geay, J., Guillot, D., Smerdon, J. E., Rajaratnam, B. (2014). Evaluating climate field reconstruction techniques using improved emulations of real-world conditions. Climate of the Past, 10(1), 1–19. http://doi.org/10.5194/cp-10-1-2014

---

## Author Comment (AC2) · 2 Mar 2017

Walter A. Perkins and Gregory J. Hakim

wperkins@uw.edu

We thank reviewer 2 for thorough and helpful comments on the manuscript.

**1   General Comments:**

The presentation of the work in the paper feels incomplete in several ways. The figures show comparisons of metrics of skill, and comparisons of estimated climate fields to a benchmark estimate, but no visualization of the reconstructions themselves, or comparisons to the actual target (the GISTEMP field and/or GMT time series)

[Figure]

We agree that the paper can be improved by the inclusion of figures detailing reconstructed data against the target data of, for example, GISTEMP. We will use the specific questions posed by you and the other reviewer to guide an expansion of the discussion/interpretation for these results.

**The authors have also neglected to describe of tabulate the computational expense associated with their reconstruction exercises. In addition, I wonder if they plan to make code for carrying out any of the reconstructions publicly available**

On our desktop workstation (4-core CPU @ 3.4 GHz), the time required for a single iteration for a 151-year reconstruction with 100 ensemble members and using 110 proxies is between 1.5 – 2 minutes. Thus, a full experiment with 1200 100-member reconstructions (100 iterations for each of the 12 blending coefficients), takes around 40 hours. However, the iterations themselves can be run in parallel on different nodes of a computing cluster. In comparison, the offline method (no forecasting) takes about 1 minute to complete a single iteration. Therefore, the addition of the LIM approximately doubles the computational expense in the worst case, but the total wall-clock time is still quite manageable for large experiments. We have not taken steps to fully optimize the code so there are likely ways to lower the overall computational expense. We plan to archive the version of the code that was used in these experiments, but also plan for a public version of the code hosted on a platform such as Github. This repository and documentation will be coordinated with other Last Millennium Reanalysis projects.

**2   Specific Comments:**

The authors would like to note that any suggestion not directly addressed will be taken into account when revising the text.

**2.1 Abstract:**

**LIMs have been shown to have comparable skill to CGCMs in what sense?**

LIMs have been shown to have comparable skill to CGCMs for forecasting surface temperature anomalies. We will be more specific in the text.

**The last sentence may need to be revised or made more specific, to address the meaning of the "dynamical evolution" to which the authors attribute improvements in skill. When I think of "dynamics," I think of the description of the underlying physical mechanisms driving changes in time, where the term is used in contrast to a "statistical" description. The LIM is purely statistical though, so I think the authors mean the term in the sense of using the model forecast as a prior for each subsequent timestep.**

By "dynamical evolution", we imply that the LIM has encoded linear dynamical properties of the system that it is calibrated on. Using a LIM to forecast the next prior imparts those linear dynamical constraints on the forecast field. We will clarify this distinction in the text.

**2.2 Introduction:**

**It seems the physical consistency issue due to use of EOFs is also a limitation of the method presented in this paper though, right? Seems a bit disingenuous to list this here as if it's a limitation the present approach will address.**

We understand the reviewer's point, but EOFs are simply used here as a basis-reduction method for forecasting, which is standard practice in the LIM literature. We make no claims that the EOFs have a physical basis in isolation (although others would make that claim). Whether the model basis is grid points, spherical harmonics, or EOFs, the goal of representing the dynamics remains the same. EOFs happen to be

a compact space that, in total, captures more variance than other approaches with similar degrees of freedom.

**The authors might expand upon what they mean by "dynamical" at the first use of the word here, to make the precise nature of their contribution more immediately accessible to a wider audience.**

We will define more clearly what we mean by "dynamics" in the context of the reconstruction.

**How many modes are retained in this study? (This detail is sufficiently important to be moved from the appendix to the main paper). What's the justification for the choice based on e-folding times of a year or greater? Are results sensitive to number of retained modes?**

We retain 8 EOFs for use in the LIM and will move this detail into the main text. In the appendix, we say that number of modes covers those with e-folding times (EFT) of 1-year or greater meaning that we picked a number of modes that encompasses those EFTs, not necessarily restricts the EFTs to 1-year or greater. For example, the CCSM4 LIM has four forecast modes with EFTs above 1-year and three modes around 1-year (EFT of 0.7 and 0.85 years respectively). Other LIMs have similar distributions of EFTs. We keep these modes because EFTs much lower than 1-year will not have a large impact when we are forecasting on annual time scales. It is worth noting that Newman (2013) keeps 11 EOFs for SSTs and 6 EOFs for land temperatures since he is using separate observational datasets, but he also states that his results are insensitive to the exact number of EOFs he retains. He also finds that most of the LIM skill on 2-9 year time scale can be reproduced with the first three forecast modes. This gives us confidence that retaining 8 EOFs for all our LIMs is a reasonable choice. We realize the statement in the text on the EOFs can be misleading and will amend it to state the reasoning behind our retained modes more clearly.

2.3   Data and experimental configuration:

**"For the prior, we used . . . . the CCSM4 last-millennium simulation": Do the authors mean this is the model used for the climatological prior used for the blending used to prevent the collapse of the ensemble as described at the end of the previous section? If so: I would expect the EOFs of the prior and the CCSM4-based LIM to be the same, but different for the other CGCM-based LIMs, thereby perhaps giving the CCSM4-based LIM an advantage, or at least somehow controlling the divergence of that ensemble differently than for the three other CGCM-based LIMs?**

Yes, we mean that the CCSM4 simulation is used as the static prior. We also considered that the CCSM4 LIM had a slight edge because of the usage of CCSM4 as a prior, but we do not believe this to be the case based on further investigation. We performed the same LIM experiments using the MPI last millennium simulation as a prior and the NOAA Merged Land-Ocean Surface Temperature analysis (MLOST) to calibrate the proxy observation models. The GMT skill between the CCSM4 and MPI-prior experiments is basically the same (with a maximum CE of 0.82), but the spatial results show the CCSM4 LIM outperforms the MPI LIM experiment in both cases (the CCSM4 spatial average CE was +0.02 above the MPI case when using the MPI prior).

**The linear observation models for proxy data" should be described in enough detail to enable reproducibility. Are the proxy data simply linear in temperature of the gridcell containing each proxy location? Or a collection of gridcells representing the regional signal of each record referred to in the next sentence?**

The observation models we use are the same as discussed in Hakim et al. (2016). The model is formed by a linear regression against the time series from the nearest grid point in the calibration dataset.

**Also, is there any particular justification for the choice of the GISTEMP product**

**for calibrating the proxy models?**

The choice of GISTEMP as the calibration data was an arbitrary choice. The observational dataset used to calibrate the proxy models will influence the reconstruction, but this sensitivity is outside the scope of the current study and is discussed in Hakim et al. (2016) (see section 4).

**Finally, it's interesting the authors use several proxy types with known differences in their spectral signatures. Is there any difference in the construction of linear observation models for the lower versus higher frequency proxies?**

All proxies we use have observations provided at annual resolutions. There is no difference in how we calibrate between them here. This is a current topic of research given that we know that some proxies provide mostly seasonal information (e.g. growing season for tree ring widths/densities).

**Optimal in what sense? What is the criterion used to determine the optimum?**

We use optimal in the sense that it can provide some improvements over the offline method in our experiments, but we see now how this is a vague usage. In our results, we use the GMT CE skill as the measure to define optimal. We then investigate the spatial results for the blending coefficient that achieved the best CE in all experiments except for the persistence forecast experiment. The persistence forecast did not have a blending coefficient showing an improvement over the offline case, so we used the best performing blending coefficient for the detrended GMT CE.

2.4   Results and Discussion:

**be more specific than writing the CRPS and CE results are "generally consistent." Do you mean the rank of models is the same as measured by both statistics? Line 18-20: Similar to preceding comment: it's imprecise to say there are "slight differences in results. . . when comparing CE and CRPS." These are**

**two different metrics that measure different things in the first place. I wonder again if the authors mean to make a statement comparing the rank of models as measured by the two different metrics?**

Yes, by generally consistent we mean that the rank of the experiments is the same, and the behavior of the two metrics across different blending coefficients is similar. We realize that the CRPS and CE are different metrics and should not be expected to be exactly the same. We simply want to bring attention to the fact that CE and CRPS are largely giving us the same information for the full GMT results. However, we also show with the detrended GMT how the CRPS can give a very different result from CE (persistence experiment for a=1.0) where it could be very misleading to a user who cares about interannual variability. We will alter this section to make the language more precise.

**In all figures, the authors show central estimates across ensembles, but no measures of uncertainty. Once it has been established in the results that the skill varies with the blending coefficient, it might be interesting to show some analysis of estimates and uncertainty across ensemble members for fixed "a" (probably at the value that optimizes one metric or another).**

We have done some uncertainty quantification, and can indeed include an expansion of this information in the main text or additionally in the supplementary information.

**I would speculate that this underestimation of trend in combination with skillful match of phase and amplitude of GMT variability might be interpreted in terms of the paleoclimate proxies as high-frequency bandpasses of the climate signal, that do not tend to preserve the low-frequency signal. This is a well-known feature of many dendrochronologies, for example, although there do exist "standardization" methodologies to prepare tree ring time series to preserve the low-frequency signal. It would be interesting to know whether the proxy time series used in this study have been prepared using methods aiming to preserve low**

**frequency climate variability.**

We used proxy records contained in the PAGES 2k Consortium (2013) database with no additional processing. The proxies in this database were selected as being representative of annual or warm season temperature variability. We do not believe it is the proxy data controlling the variation in reconstructed trends. Instead, we think the trend changes are related to the LIM forecasts and blending. This is because the same proxies are used in each experiment, and from these same proxy records we find varying trend estimates (including overestimates) that provide better skill than the offline case. The aspect controlling the trends is the relative weighting between the prior and proxy information, which is determined by the variance (uncertainty) of the two sources. The uncertainty of the proxies is fixed, so that means the weighting is being affected by the LIM forecast and the amount of blending between the static and forecast states. We discuss this on page 8 lines 4-12.

**Subsection 4.1 is missing figures and reporting of the estimated GMT time series compared to the target GMT time series. Pp. 9, line 21– seems odd not to show some spatial measures of skill against the target, rather than just against the offline case.**

Thank you for this suggestion, we will include figures of the actual temperature reconstructions and spatial measures against the GISTEMP target.

**Is there climatic significance to this North Atlantic/Barents Sea area that might explain why the LIM forecast- based reconstructions seem to improve skill there compared to the offline case? Or can the authors speculate as to why this region has low skill in the offline case to begin with to explain the near uniform improvements there under forecasting?**

The North Atlantic/Barents Sea region is a region near the sea ice edge that seems to be poorly constrained by proxy data alone. We inspected a grid point northeast of Iceland where there are negative CE values to assess the causes of the improvement. We

found that the temperature variance of the offline experiment is an order of magnitude larger than the variance in GISTEMP at this location. The CCSM4 LIM reconstruction had approximately 30% as much variance in this location as in the offline case. There was not a significant change in correlation or trend between the offline and LIM experiment, so the change in temperature variance is the primary factor in CE improvement. As far as why we might expect a LIM to help here, a modeling study used a LIM to assess the predictability of the HadCM3 coupled climate model in the North Atlantic (Hawkins and Sutton, 2009), suggesting that there is decent predictability in the North Atlantic/Barents Sea region with annual lead time (see Fig. 6 in their paper). The Newman (2013) study also shows predictability for this region when using their detrended LIM for 2-5 and 6-9 year lead times. Additionally, a preliminary LIM predictability analysis we performed shows some positive anomaly correlation skill for this region in all the LIMs we test in this study.

**"There is a clear distinction between LIMs calibrated on data from the shorter instrumental era, and the millennium-scale climate simulation data"– This is an interesting point. Remind readers explicitly at this point which are which, so that readers can easily reference what you're talking about in the figures. Also, can you describe the distinction you mean clearly and precisely? Looks to me like the millennial-scale ones have fewer regions of large-amplitude degradation in CE relative to the offline case.**

We will rework the sentence after this (pp10 line 4-5) to make this point clearer.

**3   References:**

Hawkins, E.,  Sutton, R. (2009). Decadal predictability of the Atlantic Ocean in a coupled GCM: Forecast skill and optimal perturbations using linear inverse modeling. Journal of Climate, 22(14), 3960–3978. http://doi.org/10.1175/2009JCLI2720.1

[Figure]

Hakim, G. J., Emile-Geay, J., Steig, E. J., Noone, D., Anderson, D. M., Tardif, R., . . . Perkins, W. A. (2016). The Last Millennium Climate Reanalysis Project: Framework and First Results. Journal of Geophysical Research: Atmospheres. http://doi.org/10.1002/2016JD024751

PAGES 2k Consortium. (2013). Continental-scale temperature variability during the past two millennia. Nature Geoscience, 6, 339–346. http://doi.org/10.1038/NGEO1797

Newman, M. (2013). An Empirical Benchmark for Decadal Forecasts of Global Surface Temperature Anomalies. Journal of Climate, 26(14), 5260–5269. http://doi.org/10.1175/JCLI-D-12-00590.1
* * *